



# Triple oxygen isotope composition of $CO_2$ in the upper troposphere and stratosphere

Getachew Agmuas Adnew[1], Gerbrand Koren[2], Neha Mehendale[1], Sergey Gromov[3], Maarten Krol[1,4], and Thomas Röckmann[1]

[1]Institute for Marine and Atmospheric research Utrecht (IMAU), Physics Department, Science Faculty, Utrecht University, Princetonplein 5, 3584 CC Utrecht, the Netherlands
[2]Copernicus Institute of Sustainable Development, Utrecht University, Princetonlaan 8a, 3584 CB Utrecht, the Netherlands
[3]Max Planck Institute for Chemistry, Hahn-Meitner-Weg 1, 55218 Mainz, Germany
[4]Meteorology and Air Quality, Wageningen University, Wageningen, the Netherlands

**Correspondence:** Getachew Agmuas Adnew (g.a.adnew@uu.nl)

**Abstract.** High precision measurements of the triple oxygen isotope composition of $CO_2$ ($\Delta'^{17}O$) can be used to estimate biosphere-atmosphere exchange of $CO_2$, the residence time of tropospheric $CO_2$ and stratosphere-troposphere exchange. In this study, we report measurements of the $\Delta'^{17}O(CO_2)$ from air samples collected during two aircraft based programs, CARIBIC and StratoClim. CARIBIC (Civil Aircraft for the Regular Investigation of the atmosphere based on an Instrument Container) provided air samples from numerous transcontinental flights in the upper troposphere/lower stratosphere region. StratoClim (Stratospheric and upper tropospheric processes for better climate predictions) conducted intensive campaigns with the high-altitude aircraft M55 Geophysica during the Asian Summer Monsoon Anticyclone (ASMA), providing air samples from altitudes up to 21 km.

Using high precision $\Delta'^{17}O$ measurements of the CARIBIC samples, we show that the $N_2O$-$\Delta'^{17}O$ correlation, previously observed in the stratosphere, extends to the upper troposphere. Moreover, we found no significant spatial or hemispheric differences in $\Delta'^{17}O(CO_2)$ for the upper tropospheric samples collected during the CARIBIC program. However, in many of the StratoClim samples, with significant stratospheric contributions, we observed a much lower $N_2O$-$\Delta'^{17}O$ slope compared to CARIBIC samples and previous publications. This deviation is attributed to change in eddy diffusion above the tropopause within the ASMA, confirming previously published model calculations. These samples provide the first experimental evidence that differences in vertical mixing/transport can lead to significantly different $N_2O$-$\Delta'^{17}O$ slopes. High precision $\Delta'^{17}O$ measurements can identify ejections of tropospheric air into the stratosphere based on the slope of the $N_2O$-$\Delta'^{17}O$ correlation, as both tracers have chemical lifetimes longer than their transport times.

Furthermore, we use the $\Delta'^{17}O$ measurements from the lower stratosphere and the upper troposphere to estimate global stratospheric production and surface removal of the isotope tracer $\Delta'^{17}O$. The removal estimate is then used to derive an independent estimate of global vegetation exchange of $CO_2$, confirming earlier estimates based on surface level $\Delta'^{17}O$ measurements.



# 1 Introduction

Measurements of the isotope composition of $CO_2$ ($\delta^{13}C$ and $\delta^{18}O$, see definition below) are used in many ways to understand
sources and sinks of $CO_2$ in the global carbon cycle (Farquhar et al., 1993; Welp et al., 2011; Cuntz et al., 2003b; Ciais
et al., 1997a, 1995). The flux estimates of $CO_2$ from the terrestrial biosphere are poorly constrained in current carbon cycle
models (Piao et al., 2013). By using $\delta^{13}C$ it is possible to differentiate between the ocean sink and biosphere activity. However,
distinguishing respiration, photosynthesis and combustion signals using $\delta^{13}C$ of $CO_2$ is impossible (Ciais et al., 1995). To
address this challenge, $\delta^{18}O$ of $CO_2$ is a valuable tool for distinguishing respiration from photosynthesis. The $\delta^{18}O$ value of
$CO_2$ is higher during photosynthesis than during respiration due to isotope exchange of $CO_2$ with isotopically different leaf
water and soil water pools, as leaf water typically has a higher $\delta^{18}O$ value than soil water (Ciais et al., 1997a, b; Welp, 2011;
Farquhar et al., 1993; Cuntz et al., 2003a, b; Francey and Tans, 1987; Yakir, 2020).

One of the limitations of using $\delta^{18}O$ of $CO_2$ to distinguish photosynthesis and respiration is the oxygen isotope exchange
of $CO_2$ with liquid water in surface water bodies or vegetation, since this isotope exchange affects $\delta^{18}O$ of $CO_2$, without
contributing to a net flux. Numerous equilibrium and kinetic effects can alter the $\delta^{18}O$ value of water and other molecules
(Cuntz et al., 2003a, b; Peylin et al., 1999). Importantly, the isotopic composition of leaf water at the $CO_2$-$H_2O$ exchange site
in the mesophyll is not well understood, primarily due to fractionation associated with evaporation, transport, and diffusion
(Adnew et al., 2020, 2021, 2023; Gan et al., 2002; Cousins et al., 2006; Song et al., 2015; Landais et al., 2006; Cernusak et al.,
2016; Helliker and Ehleringer, 2000). These unknown physico-chemical fractionation processes introduce large uncertainty
into estimates of gross fluxes of $CO_2$ when using $\delta^{18}O$ as quantitative tracer. This uncertainty can potentially be reduced using
the triple oxygen isotopic composition of $CO_2$, $\Delta'^{17}O$ (see equation 4), which is a combination of the $\delta^{17}O$ and $\delta^{18}O$ isotopic
composition (Hoag et al., 2005; Koren et al., 2019; Adnew et al., 2020).

Stratospheric $CO_2$ has a high $\Delta'^{17}O$ value, i.e., $\Delta'^{17}O \gg 0$, due to isotope exchange with $O(^1D)$ produced from $O_3$ photol-
ysis (Thiemens et al., 1991, 1995b; Lyons, 2001; Lämmerzahl et al., 2002; Thiemens, 2006; Kawagucci et al., 2008; Wiegel
et al., 2013; Yung et al., 1991b, 1997b; Shaheen et al., 2007). The main sink for this higher $\Delta'^{17}O$ signal of $CO_2$ is isotope
exchange with leaf, soil and ocean water at the Earth's surface, after $\Delta'^{17}O$ enriched $CO_2$ has re-entered the troposphere via
the large-scale Brewer-Dobson circulation and synoptic eddy diffusion (Boering et al., 2004; Thiemens et al., 2014; Liang and
Mahata, 2015; Francey and Tans, 1987). With the exception of stratospheric $CO_2$, $\Delta'^{17}O$ variations in nature are much smaller
compared to $\delta^{18}O$ variations, and are better defined, as conventional biogeochemical processes follow a well-defined three-
isotope fractionation slope (Barkan and Luz, 2005, 2007, 2012; Landais et al., 2006; Angert et al., 2004, 2003). Furthermore,
the triple oxygen isotope fractionation slopes for specific processes are independent of the source water isotope composition,
insensitive to temperature, and process-specific (Landais et al., 2006; Hofmann et al., 2012). As a result, $\Delta'^{17}O$ is less affected
by the numerous physico-chemical fractionation processes mentioned above, and may provide an additional constraint for



quantifying the gross fluxes of the terrestrial carbon cycle than measuring $\delta^{18}O$ alone (Koren et al., 2019; Hoag et al., 2005; Liang et al., 2017a, 2023; Hofmann et al., 2017; Adnew et al., 2020; Thiemens et al., 2014).

In addition to quantifying the gross fluxes of the terrestrial carbon cycle, $\Delta^{'17}O$ can provide useful information for concerning stratospheric intrusions (Liang and Mahata, 2015; Steur et al., 2024), stratosphere-troposphere exchange, (Boering et al., 2004; Luz et al., 1999), atmospheric transport and chemistry in the mesosphere and stratosphere (Liang et al., 2007, 2008), combustion processes (Laskar et al., 2016; Horváth et al., 2012) and for estimating the residence time of $CO_2$ in the troposphere (Liang et al., 2017b; Laskar et al., 2019; Hoag et al., 2005).

High precision measurements of $\Delta^{'17}O(CO_2)$ are particularly interesting in the tropical upper troposphere/lower most stratosphere region, which is remote from the sources and sinks. Therefore, $\Delta^{'17}O(CO_2)$ can be used to study the influence of the stratosphere-troposphere exchange on the variations and dynamics of $\Delta^{'17}O$ and its correlation with other long- and short-lived trace gases. Additionally, these measurements may be valuable for using $\Delta^{'17}O$ as a tracer to quantify gross fluxes of $CO_2$, as this region links the stratosphere, where $\Delta^{'17}O$ of $CO_2$ is produced, to the troposphere, where $\Delta^{'17}O$ of $CO_2$ is "washed out". Furthermore, it is interesting to investigate whether $\Delta^{'17}O$ varies spatially in the upper troposphere and whether it is possible to detect large scale dynamic phenomena that happen in the Asian Summer Monsoon Anticyclone (ASMA). The Monsoon circulation system has a large variability in its spatial extent and the ASMA can reach the Mediterranean, North-east Africa and East Asia (Annamalai and Slingo, 2001; Garny and Randel, 2013; Pan et al., 2016; Vogel et al., 2019).

Here we present high-precision measurements of $\Delta^{'17}O(CO_2)$ from a total of 85 air samples collected in the upper troposphere and stratosphere as part of the CARIBIC and StratoClim aircraft air sampling projects. We investigate the spatial distribution (horizontal and vertical) of $\Delta^{'17}O(CO_2)$, signals associated with the ASMA, and refine previous estimates of the net stratosphere-troposphere flux of $\Delta^{'17}O(CO_2)$ and constraints on surface $CO_2$ emissions.

## 2 Materials and Method

### 2.1 Definitions and notation

Variations in isotopic abundance are reported as deviations of a heavy-to-light isotope ratio in a sample relative to a reference material (Eq. 1). For oxygen the reference material is Vienna Standard Mean Ocean Water (VSMOW) whereas for carbon the reference material is Vienna Pee Dee Belemnite (VPDB). Since isotope variations are small, they are usually reported in per mill (‰) in delta notation.

$$\delta^x = \frac{^xR_{sam}}{^xR_{std}} - 1 \tag{1}$$

where $x$ can be 13, 17 and 18 (for $^{13}C$, $^{17}O$ and $^{18}O$, respectively). The indices *sam* and *std*, stand for sample and standard, respectively and $R$ is the ratio between the heavy isotope and the light isotope of the respective element, for instance for $^{13}C$ and $^{18}O$, $^{13}R = \frac{^{13}C}{^{12}C}$ and $^{18}R = \frac{^{18}O}{^{16}O}$, respectively. The variations in $\delta^{17}O$ and $\delta^{18}O$ are closely related during most physico-chemical fractionation processes according to Eq.2 (Matsuhisa et al., 1978; Young et al., 2002).



$$\left(\frac{^{18}R_{sam}}{^{18}R_{std}}\right)^{\theta_p} = \frac{^{17}R_{sam}}{^{17}R_{std}} \qquad (2)$$

The exponent $\theta_p$ denotes a three isotope slope that occurs from a single process. Equation 2 can be expressed in $\delta$- notation as:

$$ln(\delta^{17}O+1) - \theta_p \times ln(\delta^{18}O+1) = 0 \qquad (3)$$

The deviation of the left hand side of equation 3 from zero is defined as $\Delta'^{17}O$ (Eq. 4).

$$\Delta'^{17}O = ln(\delta^{17}+1) - \lambda_{ref} \times ln(\delta^{18}O+1) \qquad (4)$$

where the process-dependent exponent $\theta_p$ has been replaced by a three-isotope reference slope $\lambda_{ref}$. Note that the choice of $\lambda_{ref}$ is arbitrary since in nature isotopic compositions rarely reflect fractionation from a single process, but instead integrate multiple fractionating processes and several three-isotope values (Adnew, 2020; Barkan and Luz, 2005, 2007, 2012; Angert et al., 2004). For most natural processes $\lambda_{ref}$ ranges from 0.5 to 0.5305 (Thiemens, 2006; Matsuhisa et al., 1978; Young et al.,

2002; Cao and Liu, 2011; Thiemens et al., 1991; Kaiser, 2008) with some exceptions (Adnew et al., 2022; Hayles et al., 2017; Hayles and Killingsworth, 2022). In this study we use a $\lambda_{ref}$ value of 0.528, the value associated with meteoric water (Meijer and Li, 1998; Luz and Barkan, 2010).

## 2.2 Calculation of net isotope flux from stratosphere to troposphere

As described in detail by Plumb and Ko (1992) and Plumb (2007), gases that are chemically long-lived relative to vertical and

quasi-horizontal transport time scales exhibit compact correlations in the stratosphere, and the slope of the observed correlation between the two tracers is equal to the ratio of their net vertical fluxes. Following this approach, Luz et al. (1999) and Boering et al. (2004) determined the global annual mean net isotope flux (NIF) from the stratosphere (ST) to the troposphere (T) for $\Delta'^{17}O$ of $CO_2$ using the $\Delta'^{17}O(CO_2)$-$N_2O$ correlation as described in detail by Garofalo et al. (2019).

$$\Delta'^{17}O(CO_2) - NIF = MF \times [CO_2]_{ST} \times \left[m \times \frac{-L_{N_2O}}{MF} + \Delta'^{17}O(CO_2)_T\right] - MF \times [CO_2]_T \times \Delta'^{17}O(CO_2)_T \qquad (5)$$

The terms $MF$, $L_{N_2O}$, and $m$ stand for the total air mass flux from the stratosphere to the troposphere, the global $N_2O$ loss rate and the correlation slope between $\Delta'^{17}O$, respectively. In our analysis, this correlation slope is determined using a Williamson-York bivariate fit, accounting for uncertainties in both the $\Delta'^{17}O(CO_2)$ and $N_2O$ data (Mikkonen et al., 2019). There is a significant variation in estimates for the air mass flux from the stratosphere to the troposphere. For instance, the estimates by Holton et al. (1995) and Appenzeller et al. (1996) differ by a factor of three in their calculated cross-tropopause

air mass fluxes, but note that the troposphere definitions are also different. As described in Garofalo et al. (2019), the net isotope flux is not strongly sensitive to the actual air mass fluxes. Since the mole fraction of $CO_2$ in the lower stratosphere and troposphere is the same within about 1 %, equation 5 can be simplified to:



$$\Delta^{'17}O(CO_2) - NIF = -m \times L_{N_2O} \times [CO_2] \tag{6}$$

The uncertainty in the estimated net annual mean flux of $\Delta^{'17}O(CO_2)$ depends on the uncertainty in depends on the uncertainty
of $m$ and $L_{N_2O}$. In our calculation, we used 370 ppm for $[CO_2]$, the average value for CARIBIC samples, most of them collected
in the year 2001.

## 2.3   Estimating gross surface flux using $\Delta^{'17}O(CO_2)$

The CARIBIC samples analysed in this study cover latitudes from 30° south to 54° north (see Figure 1), we used these samples
to estimate surface emissions using a mass balance box model calculation. The box model used in this study is not suitable to
simulate the spatial and temporal variability of $\Delta^{'17}O(CO_2)$ (Koren et al., 2019). Our box model is an extended version of those
used by Hoag et al. (2005) and Liang et al. (2017b), incorporating an intermediate upper troposphere (*UT*) box. Additionally,
our study includes $\Delta^{'17}O$ measurements from both hemispheres and different seasons. The $\Delta^{'17}O(CO_2)$ value in the *UT*
depends on the $\Delta^{'17}O(CO_2)$ value of stratosphere (*ST*) and surface processes (assimilation/photosynthesis (*A*), respiration (*R*),
soil invasion (*SI*), ocean (*o*) and anthropogenic emission (*anth*)) and the corresponding fluxes (*F*). At steady state, the mass
balance equation for $\Delta^{'17}O(CO_2)$ in the upper troposphere is:

$$F_A \times \Delta_A\Delta^{'17}O + F_{anth} \times \Delta^{'17}O_{anth} + F_{ST} \times \left(\Delta^{'17}O_{ST} - \Delta^{'17}O_{UT}\right) + F_o \times \left(\Delta^{'17}O_o - \Delta^{'17}O_{UT}\right) +$$
$$F_R \times \left(\Delta^{'17}O_R - \Delta^{'17}O_{UT}\right) + F_{SI} \times \left(\Delta^{'17}O_{SI} - \Delta^{'17}O_{UT}\right) = 0 \tag{7}$$

where $F_A = F_{al} - F_{la} = 0.88 \times GPP$ (Ciais et al., 1997a), with $al$, $la$, and $GPP$ being atmosphere-to-leaf flux, leaf-to-
atmosphere flux, and gross primary production, respectively. $\Delta_A\Delta^{'17}O$ is the discrimination against $\Delta^{'17}O$ during assimi-
lation, and calculated as $\Delta_A\Delta^{'17}O = (\Delta^{'17}O_{UT} - \Delta^{'17}O_M) \times (-0.150 \times e^{3.707 \times \frac{c_m}{c_a}} + 0.028)$ as described by Adnew et al.
(2020) (see the supplementary material for more detail). This parameterization was derived from leaf cuvette studies using
both $C_3$ and $C_4$ plants under different light conditions. $c_m$ and $c_a$ represent the $CO_2$ mole fraction in the mesophyll and the
atmosphere, respectively. $\Delta^{'17}O_M$ is the $\Delta^{'17}O$ of $CO_2$ in the mesophyll, calculated from the $\Delta^{'17}O$ value of meteoric wa-
ter (MW) (Landais et al., 2007; Barkan and Luz, 2012; Bottinga and Craig, 1968; Brenninkmeijer et al., 1983), see also the
supplementary material.

The estimates of soil invasion fluxes are highly uncertain, the reported values in the literature vary from < 10 PgC/yr (Stern
et al., 2001) to 450 PgC/yr (Wingate et al., 2009). For this study, we assume that $F_{SI}$ and $F_R$ are equal, and $F_A = 0.88 \times GPP = 0.88 \times (NEP - F_R)$ (Liang et al., 2017b). Substituting this term in equation 7:

$$F_A \times \Delta_A\Delta^{'17}O + F_{anth} \times \Delta^{'17}O_{anth} + F_{ST} \times \left(\Delta^{'17}O_{ST} - \Delta^{'17}O_{UT}\right) + F_o \times \left(\Delta^{'17}O_o - \Delta^{'17}O_{UT}\right) +$$
$$\left(\frac{F_A}{0.88} - NEP\right) \times \left(\Delta^{'17}O_R - \Delta^{'17}O_{UT}\right) + \left(\frac{F_A}{0.88} - NEP\right) \times \left(\Delta^{'17}O_{SI} - \Delta^{'17}O_{UT}\right) = 0 \tag{8}$$



**Table 1.** Summary of parameters used for the box model mass balance calculation in Equation 8. $T$ and $RH$ represents temperature in Kelvin and relative humidity near the Earth's surface, respectively.

| Parameter | value or description | ref |
|---|---|---|
| $\alpha^{18}_{trans}$ | $1.002644 - \frac{3.206}{T} + \frac{1534}{T^2}$ | Bottinga and Craig (1968) |
| $\alpha^{18}_{CO_2-H_2O}$ | $\frac{17.604}{T} + 0.98207$ | Brenninkmeijer et al. (1983) |
| $\alpha^{18}_{diff-water}$ | $1.0016\,‰$ | Farquhar and Lloyd (1993) |
| $\alpha^{18}_{diff-soil}$ | $1.0072\,‰$ | Miller et al. (1999) |
| $\theta_{trans}$ | $0.522 - 0.008 \times RH$ | Landais et al. (2007) |
| $\theta_{CO_2-H_2O}$ | $0.5229$ | Barkan and Luz (2012) |
| $\theta_{diff-H_2O}$ | $0.50$ | calculated |
| $\theta_{diff-air}$ | $0.509$ | Young et al. (2002) |
| GPP | Gross primary production | calculated |
| $F_A$ | $0.88\times$ GPP | Ciais et al. (1997b) |
| $F_{al}$ | $\frac{c_a}{c_a-c_m} \times F_A$ | Farquhar et al. (1993) |
| $F_{la}$ | $\frac{c_m}{c_a-c_m} \times F_A$ | Farquhar et al. (1993) |
| $\frac{c_m}{c_a}$ | $0.65$ | Farquhar et al. (1993) |
| $M_a$ | $807 \pm 6$ PgC | Stocker et al. (2013) |
| $F_o$ | $80 \pm 6$ PgC/yr | Stocker et al. (2013) |
| $F_{anth}$ | $9 \pm 0.8$ PgC/yr | Stocker et al. (2013) |
| NEP | $10$ PgC/yr | Saugier et al. (2001) |
| $\Delta'^{17}O_{MW}$ | $0.033 \pm 0.005\,‰$ | Barkan and Luz (2012) |
| $\theta_{deg-equ}$ | $1.0$ | assumed |
| $\Delta'^{17}O_{OW}$ | $-0.005 \pm 0.001\,‰$ | Barkan and Luz (2012) |

The $\Delta'^{17}O_{anth}$ value is $-0.446 \pm 0.077\,‰$ (Horváth et al., 2012; Laskar et al., 2016). The $\Delta'^{17}O_o$ value is calculated from

the $\Delta'^{17}O$ value of ocean water (OW) (Table 1). Similarly, the $\Delta'^{17}O$ values for soil invasion and respiration are derived from the $\Delta'^{17}O$ value of meteoric water (Liang et al., 2023; Koren et al., 2019; Hofmann et al., 2017). For the calculations, we used a surface temperature of 15°C, and a relative humidity of 75 % (Dai, 2006). The sensitivity of $\Delta_A\Delta'^{17}O$ to temperature and relative humidity is shown in Figure S1 of the supplementary material. The value of $\Delta_A\Delta'^{17}O$ increases with an increase in relative humidity but decreases with increasing temperature. Using the parameters described above and provided in Table

1 and Equation 8, we estimated the gross primary production (GPP), surface flux (including land and ocean) and the oxygen isotope residence time (turnover time) of $CO_2$ in the atmosphere. The oxygen isotope residence time of $CO_2$ in the atmosphere is defined by the ratio of the atmospheric $CO_2$ mass loading ($M_a$) and the $CO_2$ surface flux (Welp et al., 2011). The error for all estimated values is determined using a Monte Carlo simulation using $10^6$ runs, where values of the input parameter are randomly picked from an input value distribution of these parameters defined by their uncertainty.



The sensitivity of GPP to relative humidity, temperature, the $\Delta^{'17}O$ value of the upper troposphere, and the net flux of
$\Delta^{'17}O$ from the stratosphere to the troposphere is shown in Figures S2 and S3 of the supplementary material. A higher net
flux of $\Delta^{'17}O$ to the troposphere leads to an increase in GPP to fulfill the isotope balance of equation 8. An increase in relative
humidity leads to a decrease in GPP due to an increase in $\Delta_A\Delta^{'17}O$. Conversely, an increase in temperature results in an
increase in GPP, as higher temperatures cause a decrease in $\Delta_A\Delta^{'17}O$, as described above. The parameters used in the mass
balance calculations and their errors are provided in Table 1.

### 2.4 De-trending and classification of samples into upper troposphere and stratosphere

For comparison with previously published data and between the CARIBIC and StratoClim samples measured in this study, all
$N_2O$ measurements were trend-corrected to the year 2001 following a similar approach described in Koren et al. (2019).

$$N_2O_{det} = N_2O_{obs} \times \left[1 - \frac{N_2O_{growthrate}}{N_2O_{ref}} \times (t_{ref} - t_{obs})\right]^{-1} \tag{9}$$

Here, $N_2O_{obs}$, $N_2O_{det}$, $N_2O_{ref}$ and $N_2O_{growthrate}$ refer to the observed and detrended mole fractions of $N_2O$, the mole
fraction of $N_2O$ at the reference time and the growth rate of $N_2O$ in the troposphere, respectively. The variables $t_{obs}$ and $t_{ref}$
represent the time of observation and the reference time (1 July 2001). The growth rate of $N_2O$ in the troposphere, used in this
study, is 0.75 ppb/year (Stocker et al., 2013). The mole fraction of $N_2O$ at the reference time (1 July 2001) is 316.24 ppb (Lan
et al., 2024).

The mole fraction of $CH_4$ in the StratoClim data was detrended to 2007 using an average growth rate of 12 ppb/year (Nisbet
et al., 2019). The 2007 mole fraction of $CH_4$ is assumed to be the same as the $CH_4$ mole fraction in 2001, since the atmospheric
$CH_4$ mole fraction was stable between 2000 and 2007 (Nisbet et al., 2019).

We used the $N_2O$-CO correlation to classify air samples into upper troposphere and stratospheric samples as described in
detail by Assonov et al. (2013). The CARIBIC samples are mostly from the upper troposphere (UT) and include a few from the
lower stratosphere (see Figure S4 in the supplementary material). Based on the mole fraction of $N_2O$, we grouped the $\Delta^{'17}O$
values into two categories: upper troposphere (UT) samples ($N_2O \geq 313.5$ ppb) and the lower stratosphere (LS) samples
($N_2O < 313.5$ ppb). We further divided the lower stratosphere into two subgroups: $313.5 > N_2O > 306$ ppb and $N_2O < 306$
ppb. Furthermore, we used a zonal average tropopause pressure simulated using the TM5 model as outlined in Krol et al.
(2018), and the *age of air* tracer, to support the classification of samples into upper tropospheric and stratospheric. Most of the
samples classified as upper tropospheric based on the mole fraction of $N_2O$ were collected far below the tropopause whereas
the samples classified as stratospheric were collected above or close to the tropopause with a few exceptions (see Figure S5 in
the supplementary material).

### 2.5 Sampling

Air samples were collected from the upper troposphere and stratosphere during two different international projects, CARIBIC
and StratoClim, which are shortly described below. Figure 1 shows the geographical coordinates of the sampling locations for
the samples measured in this study.



### 2.5.1 CARIBIC samples

In the CARIBIC project (Civil Aircraft for the Regular Investigation of the atmosphere Based on an Instrument Container, CARIBIC, https://www.caribic-atmospheric.com/), samples were collected using a Boeing 767 aircraft (LTU, Germany). The flights operated at typical commercial cruising altitude between 9 and 12 km, i.e., which is the upper troposphere-lower stratosphere region at higher latitudes and the upper troposphere region in the mid latitudes and tropics (Assonov et al., 2010). The details of the CARIBIC instrument container are described in Brenninkmeijer et al. (1999). The payload of the flights before 2003 included large stainless steel canisters for collecting whole air samples (WAS) (sample size $\approx$ 340 L STP) at several locations along the flight path. On a single flight 12 discrete samples were collected. Each sample collection took about 20 min which corresponds to a horizontal distance of 250 km (Brenninkmeijer et al., 1999). The samples collected are relatively dry since the ambient temperature was always below -30 °C and the relative humidity was about 0.44 % at 20 °C (Assonov et al., 2009b). The air samples were processed soon after they returned to the Division of Atmospheric Chemistry, Max Planck Institute for Chemistry, Mainz, Germany (Pupek et al., 2005). The processing included the extraction of $CO_2$ and CO for isotope analysis, including radiocarbon (Brenninkmeijer, 1993; Assonov et al., 2009b) as well as measurement of several other trace gases including $N_2O$, $CH_4$ and $SF_6$ (Brenninkmeijer et al., 1999). The CARIBIC container was also equipped with an automated in-situ analyzer for $O_3$ (UV absorption), CO (gas chromatography (GC)) and other parameters (Brenninkmeijer et al., 1999). In this study we measured the $\Delta'^{17}O$ value of 50 CARIBIC samples.

### 2.5.2 StratoClim samples

35 additional air samples were collected on the high-altitude M55 Geophysica aircraft during two campaigns of the StratoClim project (https://www.stratoclim.org/) (Stefanutti et al., 1999; Cairo et al., 2010) in 2016 and 2017. In 2016, three flights were conducted over the Mediterranean region from Kalamata, Greece (37°2′N and 22°7′E) between August 30 and 6 September and eight flights were conducted in 2017 over the Indian subcontinent from Kathmandu, Nepal (27°46′N and 85°16′E) between 27 July and 10 August. The whole air sampler of the Institute for Marine and Atmospheric research Utrecht (IMAU), Utrecht University was used to compress air into 2 L pre-evacuated stainless-steel canisters (Kaiser et al., 2006). We analysed 16 samples from the 2016 flights and 19 samples from the 2017 flights (Table S3). Further details on sample collection using the Geophysica aircraft can be found in Kaiser et al. (2006) and Stefanutti et al. (1999).

### 2.6 Extraction of $CO_2$

For the StratoClim samples, the $CO_2$ was extracted from the whole air samples using a cryogenic extraction system developed at Utrecht University (Adnew et al., 2020, 2023). The extraction system was made of electropolished stainless steel and has four traps. The first two traps remove moisture and condensable organics at dry ice temperature whereas the third and fourths traps were used to collect $CO_2$ at liquid nitrogen temperature. The extraction was performed at a flowrate of 55 mL/min.

For the CARIBIC samples, the $CO_2$ had previously been extracted at the Max Planck Institute for Chemistry, Mainz, as described in Assonov et al. (2009b) and Pupek et al. (2005). Two "Russian Doll" cryogenic traps (Brenninkmeijer and Röckmann,



**Figure 1.** Geographical location of the CARIBIC (black markers), StratoClim2016 (blue circles) and StratoClim2017 (blue stars) air samples measured in this study. Upper tropospheric CARIBIC samples are categorized based on the geographical location where they are collected. These categories include Southern Hemisphere samples, Northern Hemisphere samples, African samples (longitude between 7° and 45°), Asian samples (longitude > 45°), and American samples (longitude < 0°).

1996; Brenninkmeijer, 1991) immersed in liquid nitrogen trapped all condensable gases including $CO_2$, $N_2O$, $H_2O$ and most

organics. After pumping out non-condesables, the traps were slowly heated and the evolving $CO_2$ and $N_2O$ was trapped in a U-trap cooled with liquid nitrogen while the $H_2O$ remained trapped in the Russian Doll Traps. The collected $CO_2$ was further dried using $P_2O_5$ and flame-sealed in clean vials made of borosolicate glass (Assonov et al., 2009b; Pupek et al., 2005).



## 2.7 Measurement of $\delta^{13}$C, $\delta^{18}$O and $\Delta'^{17}$O of CO$_2$

The $\delta^{18}$O and $\delta^{13}$C values presented below were measured using a Delta$^{PLUS}$ V isotope ratio mass spectrometer (Ther-
moScientific, Germany) in dual inlet mode at Utrehct University. The interference of N$_2$O was corrected using: $\delta^{18}$O $=$
$\delta^{18}$O$_{Measured}$ $+ \frac{[N_2O]}{[CO_2]}(\frac{347}{1000})$ and $\delta^{13}$C $= \delta^{13}$C$_{Measured}$ $+ \frac{[N_2O]}{[CO_2]}(\frac{250}{1000})$(Friedli and Siegenthaler, 1988; Sirignano et al., 2004).
Figure 2 shows a comparison of the the $\delta^{18}$O and $\delta^{13}$C of CO$_2$ of CARIBIC samples measured in this study with those previ-
ously measured at the Max-Planck-Institute for Chemistry and reported in Assonov et al. (2009b, a, 2010).

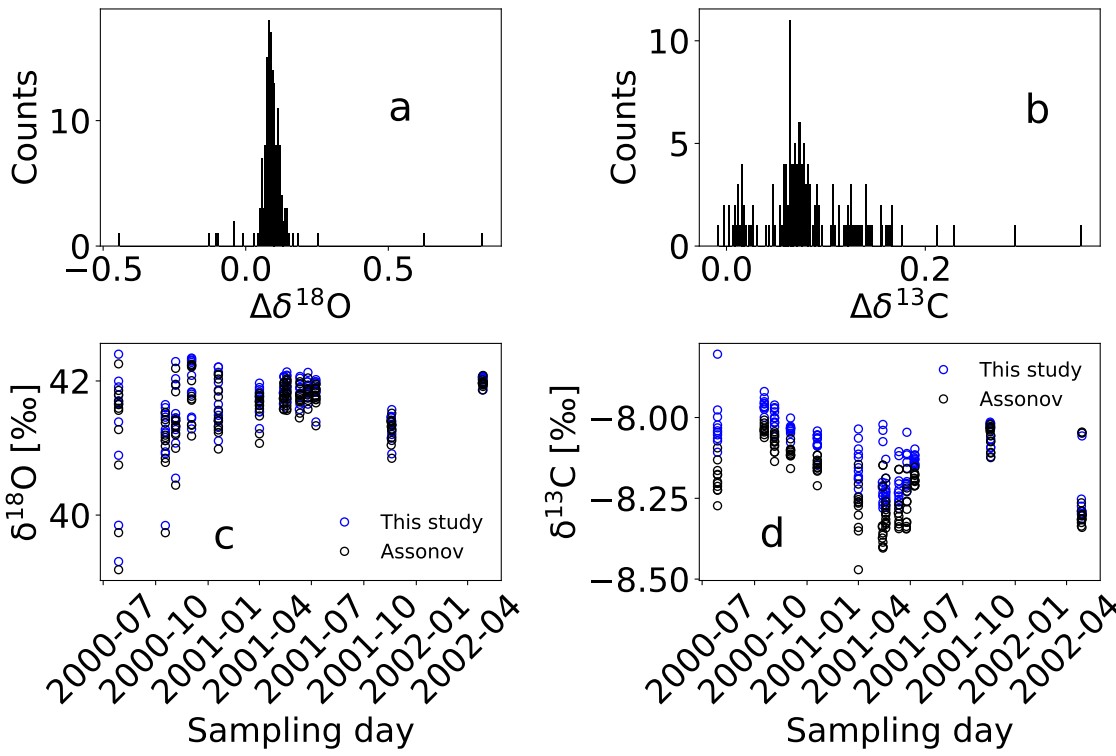

**Figure 2.** The $\delta^{18}$O and $\delta^{13}$C of CARIBIC samples and comparison with values published in Assonov et al. (2009b, a, 2010).

The $\delta^{18}$O values measured in this study are systematically higher compared to the measurement by Assonov et al. (2010)
by $0.096 \pm 0.008$ ‰ . For $\delta^{13}$C values, our measurement is also higher by $0.089 \pm 0.011$ ‰ . The reported error is the 95
% confidence limit (standard error of the mean multiplied by Student's t-factor). The tight distribution of differences is of
the order of the measurement precision, which suggests that the isotopic composition of the samples remained stable during
long-term storage in borosilicate glass vials and that the mean offset is due to scale differences. These observed offsets are
consistent with the scale uncertainty reported by Assonov et al. (2009a) and better than typical inter-laboratory uncertainties
(Levin et al., 2007).



The $\Delta'^{17}O$ of the $CO_2$ was determined using the $CO_2$-$O_2$ exchange method (Adnew et al., 2019, 2022, 2023; Barkan et al., 2015) at Utrecht University. Equal amounts of $CO_2$ sample and the laboratory reference $O_2$ (with known isotopic composition) were allowed to exchange isotopes for 2 hours in a quartz reactor at 750°C in the presence of platinum sponge at the bottom of the reactor. After the reaction, the mixture was passed through a liquid nitrogen-cooled trap to condense the

$CO_2$ and the $O_2$ was collected in a separate trap on 3 pellets of 5Å molecular sieve (1.6mm, Sigma-Aldrich, USA) at liquid nitrogen temperature. The $O_2$ was then transferred to the bellows of the dual inlet system of a $Delta^{PLUS}$ V isotope ratio mass spectrometer (ThermoFisher Scientific, Germany) and measured for its isotope composition. The $\Delta'^{17}O$ value of the original $CO_2$ was calculated from the change in the isotopic composition of the non-reacted and reacted $O_2$, and knowledge of the precise steady-state $O_2$-$CO_2$ isotope fractionations (Adnew et al., 2019; Barkan et al., 2015). The precision of the $CO_2$ isotope

measurements was 0.007 ‰ , 0.03 ‰ and 0.008 ‰ for $\delta^{13}C$, $\delta^{18}O$ and $\Delta'^{17}O$ respectively (Adnew et al., 2019, 2020, 2023).

## 3   Results

Figure 3a shows $\Delta'^{17}O(CO_2)$ as a function of latitude for the CARIBIC samples. The samples classified as upper tropospheric, $\Delta'^{17}O(CO_2)$ do not show geographic variations between large regions referred to as America, Asia and Africa, respectively (see Figure 1) and no statistically significant difference in $\Delta'^{17}O(CO_2)$ between the hemispheres (see Figure 3b). Table S2 in

the supplementary material presents the mole fraction of $CO_2$, $O_3$, $CO$, $N_2O$ and $CH_4$, along with the isotopic composition of $CO_2$ ($\delta^{13}C$, $\delta^{18}O$ and $\Delta'^{17}O$) for the analyzed CARIBIC samples. Similarly, the mole fraction of $CO_2$ and other trace gases ($O_3$, $CO$, $N_2O$ and $CH_4$), along with the isotopic composition of $CO_2$ for StratoClim samples is shown in Table S3 of the supplementary material.

Figure 4 shows the correlation of $\Delta'^{17}O(CH_2)$ measurements with the mole fractions of $CH_4$ and $N_2O$. For both sets of

samples (CARIBIC and StratoClim), $N_2O$ and $\Delta'^{17}O(CO_2)$ are clearly correlated with $R^2$ values of $\geq 0.9$ as shown in Figure 4. Similar tight correlations between $N_2O$ and $\Delta'^{17}O(CO_2)$ were reported in the previous studies as shown in Figure 5. Interestingly, StratoClim samples plot in two distinct groups of $N_2O$-$\Delta'^{17}O(CO_2)$ correlations. Most of the StratoClim samples with $\Delta'^{17}O < 0.2$ ‰ have a similar $N_2O$-$\Delta'^{17}O$ slope as the CARIBIC samples as shown in Figure 4a. However, unexpectedly the $N_2O$-$\Delta'^{17}O$ correlation slope for most of StratoClim samples with $\Delta'^{17}O > 0.2$ ‰ is much lower (-0.017 ‰ per ppb) (see

Figure 4a).

As shown in Figure 4b, there is a strong inverse correlation between $CH_4$ and $\Delta'^{17}O$. For the StratoClim samples, the correlation is higher compared to the CARIBIC samples, with $R^2$ values of $\geq 0.9$ and 0.79, respectively. For both CARIBIC and StratoClim samples with $\Delta'^{17}O < -0.2$ ‰ , no clear correlation between $CH_4$ and $\Delta'^{17}O$ can be established (see Figure 4b).

Figure 5 shows a comparison of our CARIBIC and StratoClim samples measured in this study with previously published $\Delta'^{17}O$ measurements of upper-tropospheric and stratospheric $CO_2$ (Thiemens et al., 1995a, c; Alexander et al., 2001; Lämmerzahl et al., 2002; Boering et al., 2004; Wiegel et al., 2013; Kawagucci et al., 2005; Yeung et al., 2009). Four different characteristics are shown: The three-isotope plot (Figure 5a), $\Delta'^{17}O$ as a function of altitude (only for StratoClim samples)





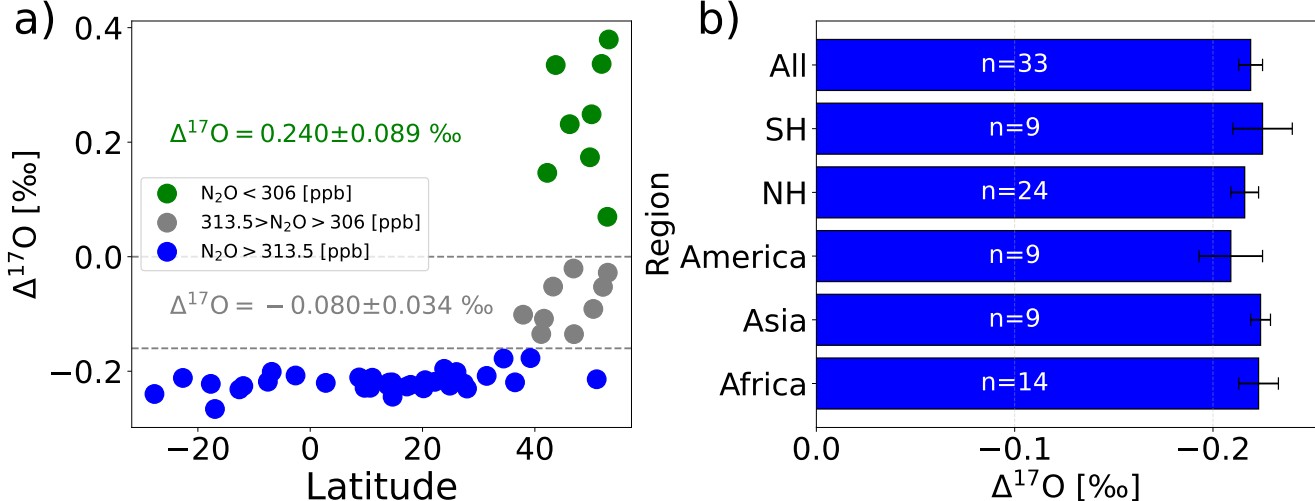

**Figure 3.** a) $\Delta'^{17}O(CO_2)$ values of CARIBIC samples as a function of latitude. The $\Delta'^{17}O$ values are grouped into three categories based on the mole fraction of $N_2O$ as described in the text. b) average $\Delta'^{17}O$ value of all upper tropospheric CARIBIC air samples and larger geographical regions. The errors are the standard error of the mean multiplied by the student t-factor for the 95 % confidence limit. *n*, *SH* and *NH* stands for the number of samples, Southern Hemisphere and Northern Hemisphere, respectively.

(Figure 5b), the $N_2O$-$\Delta'^{17}O$ correlation (Figure 5c) and the $CH_4$-$\Delta'^{17}O$ correlation (Figure 5d). In these overview plots of

$N_2O$-$\Delta'^{17}O$ and $CH_4$-$\Delta'^{17}O$ plots, the new CARIBIC and StratoClim measurements are almost indistinguishable from the measurements reported in the literature. However, when we zoom in and compare the correlation slopes, clear differences for some of the measurements become apparent such as shown in Figure 4a for the variation in the $N_2O$-$\Delta'^{17}O$ correlation between CARIBIC and StratoClim samples.

Figure 6 shows the three-isotope plot for both CARIBIC and StratoClim samples. The three-isotope plot of upper-tropospheric

samples shows a tight correlation for both CARIBIC and StratoClim samples with slopes of $0.540 \pm 0.005$ and $0.556 \pm 0.012$, respectively (Figure 6). For the StratoClim samples with higher stratospheric influence the $ln(\delta^{17}O+1)$ vs $ln(\delta^{18}O+1)$ is poorly correlated. It is apparent in Figure 6a that the vertical offsets from the fit line defined by the upper tropospheric samples get larger for larger stratospheric age, which is shown as colour coding. Figure 6b shows the correlation between the age of the air and $\Delta'^{17}O(CO_2)$. An increase in the age of the air correlates well with a progressive enrichment of $\Delta'^{17}O(CO_2)$.

**4   Discussion**

When the CARIBIC and StratoClim samples presented here are separated into upper tropospheric and stratospheric based on $N_2O$ levels, these two groups exhibit distinct characteristics. The upper tropospheric samples show a tight correlation on the three-isotope plot, whereas the stratospheric samples display significant variability without a clear correlation. Interestingly,





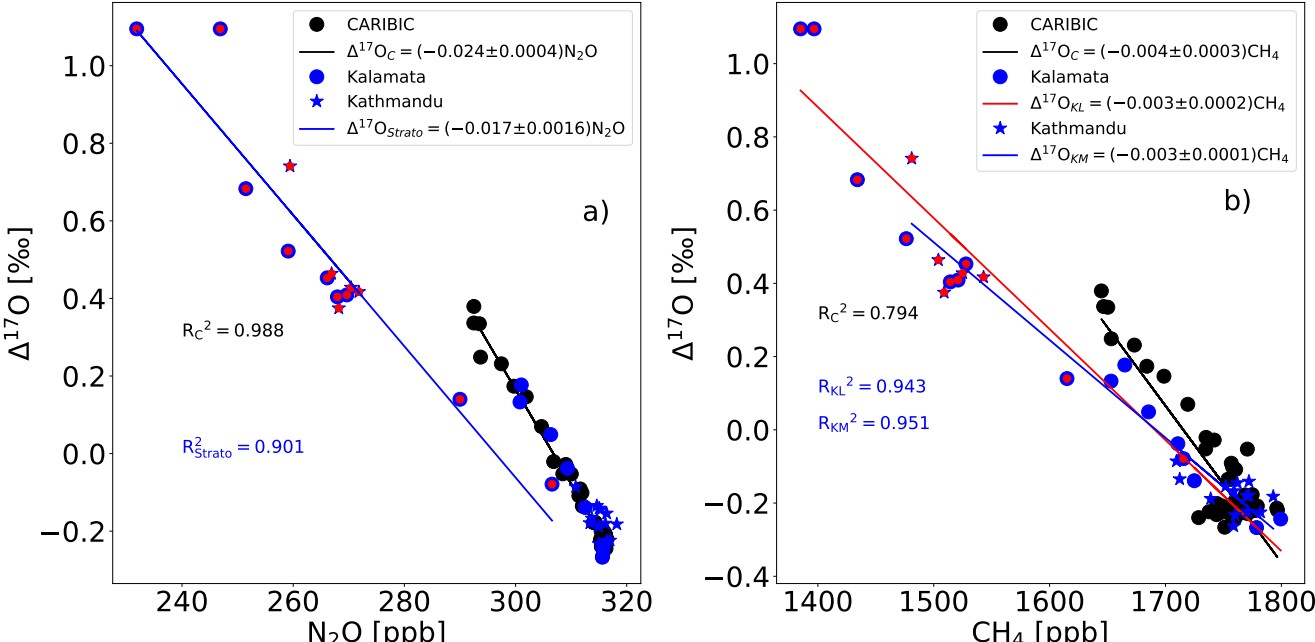

**Figure 4.** Correlation of $\Delta'^{17}O(CO_2)$ with a) $N_2O$ for StratoClim samples (StratoClim-2016 is Kathmandu (KT, blue stars) and StratoClim-2017 is Kalamata (KL, blue circles)) and CARIBIC samples (black circles) and b) $CH_4$. The errors of the linear regression slopes are the 95 % confidence interval. The data points marked in red indicate StratoClim samples with a low $N_2O$-$\Delta^{17}O$ correlation slope.

the three-isotope slopes (from a linear fit to the data) for the upper tropospheric samples are higher than the expected canonical

range for tropospheric samples ($\lambda > 0.5309$). This indicates that upper tropospheric air already has a stratospheric influence. The contribution of stratospheric air in the CARIBIC samples is evident from an excellent correlation between $\Delta'^{17}O(CO_2)$ and both $^{14}C(CO)$ and $O_3$, as illustrated in Figure 7. Both $^{14}CO$ and $O_3$ are reliable tracers for stratospheric air (Brenninkmeijer et al., 1995). Similar correlations between $^{14}CO$ and $\Delta'^{17}O(CO_2)$ have been previously reported for stratospheric $CO_2$ (Thiemens et al., 1995c; Brenninkmeijer et al., 1995).

The observed scatter in the three-isotope plot for the stratospheric samples is conceptually similar to the scatter reported in previous studies (Boering et al., 2004; Wiegel et al., 2013; Mrozek et al., 2017; Thiemens et al., 1995a). In these publications it was mostly assumed to be due to the low precision of the measurement method. However, in this study, with high precision measurements, a similar scatter in the three-isotope plot is observed, demonstrating that the scatter observed in the three-isotope plot for stratospheric samples is a real signal of mixing, transport and production of $\Delta'^{17}O(CO_2)$ (i.e. exchange between $O(^1D)$

and $CO_2$ ). The tight correlation with the age of air provides a clear clue to explain this lack of correlation. The longer $CO_2$ has been exposed to the stratospheric $CO_2$-$O(^1D)$ exchange, the higher the $\Delta'^{17}O(CO_2)$ signal is (Yung et al., 1997a; Gamo et al., 1995; Yung et al., 1991a). The fact that the different samples do not line up on a single three isotope correlation line shows that we have sampled stratospheric air originating from different upper tropospheric "entry values" before entering



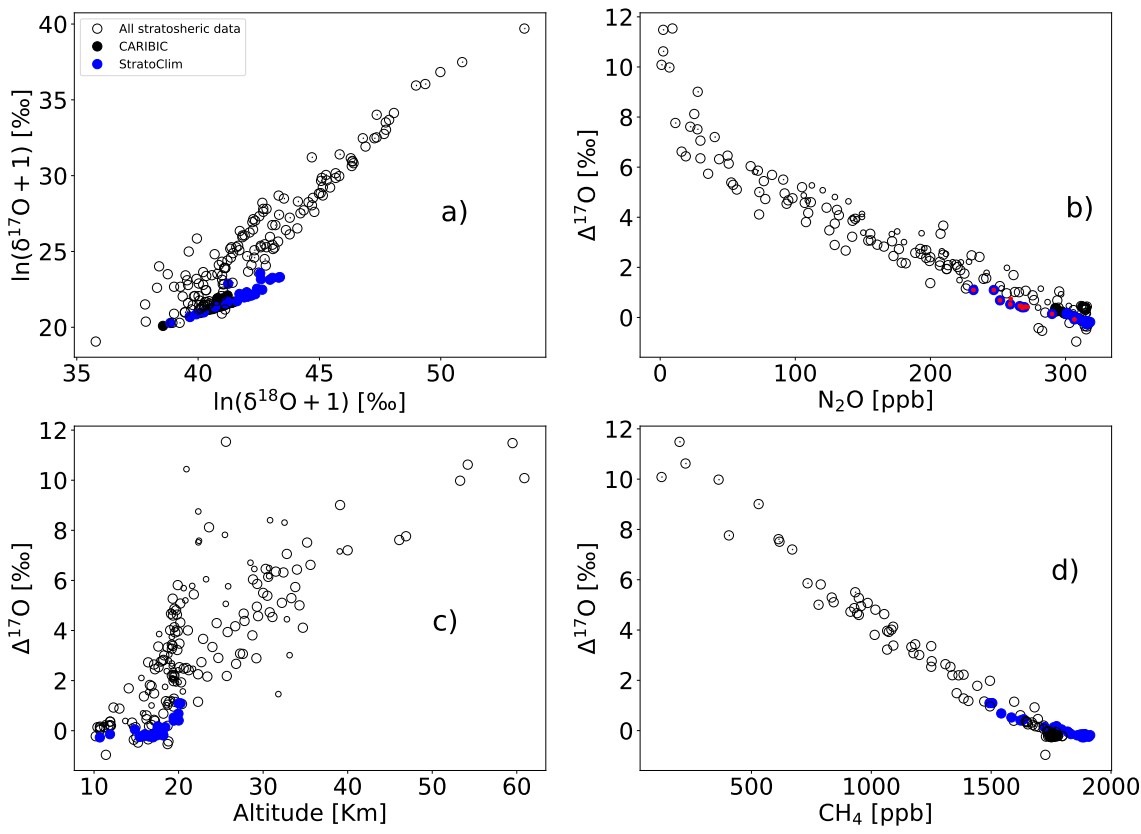

**Figure 5.** Comparison of results from CARIBIC samples (black circles) and StratoClim samples (blue circles) to previous publications (open circles). a) three isotope plot, b) correlation of $\Delta'^{17}O(CO_2)$ with $N_2O$. The data points marked in red indicate StratoClim samples with a low $N_2O$-$\Delta^{17}O$ correlation slope. c) correlation of $\Delta'^{17}O(CO_2)$ with altitude and d) correlation of $\Delta'^{17}O(CO_2)$ with $CH_4$. Previous stratospheric samples are from (Thiemens et al., 1995a, c; Alexander et al., 2001; Lämmerzahl et al., 2002; Boering et al., 2004; Wiegel et al., 2013; Kawagucci et al., 2005; Yeung et al., 2009)

the stratosphere. These differences in entry values are likely driven by the seasonality of $\delta^{17}O(CO_2)$ and $\delta^{18}O(CO_2)$. Thus,
different tropospheric air masses have entered the stratosphere from various points along the correlation line established by the upper tropospheric samples. Once in the stratosphere, photochemical isotope exchange with $O^1D$ occurs, and the samples progressively acquire a higher $\Delta'^{17}O(CO_2)$ signature as the sampled air resided longer in the stratosphere. This is clearly demonstrated by the strong correlation between $\Delta'^{17}O(CO_2)$ and the age of air, and it was not clearly visible in previous studies because of the higher measurement uncertainty.





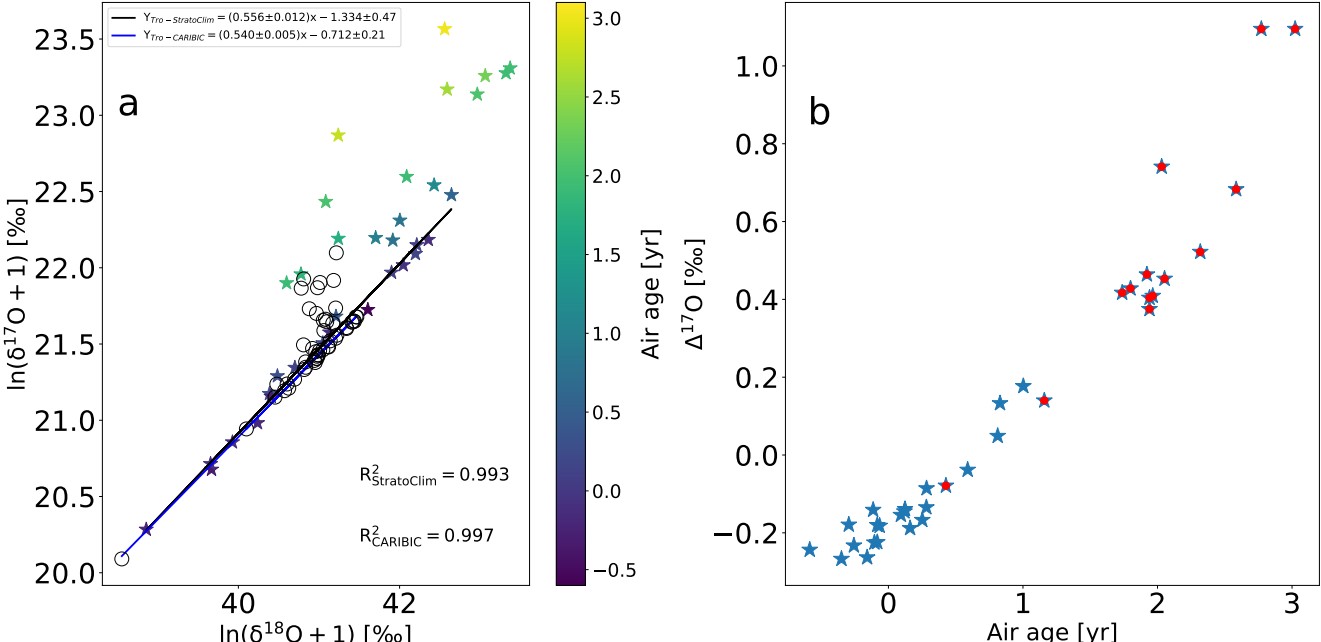

**Figure 6.** a) Three isotope plot for StratoClim (stars) and CARIBIC (open circles) samples. The linear-regression is a fit to the samples identified as upper tropospheric based on the $N_2O$ mole fraction. The color bar indicates the age of the air mass for the StratoClim samples reported in Adcock et al. (2021). The stated errors for the linear regression parameters are the 95 % confidence intervals. b) Relationship between the age of the air and $\Delta'^{17}O(CO_2)$ for StratoClim samples. The data points marked in red indicate samples with a low $N_2O$-$\Delta^{17}O$ correlation slope.

Interestingly, despite our high measurement precision, we did not observe significant geographic variations in $\Delta'^{17}O(CO_2)$ of CARIBIC air samples classified as upper troposphere (see Figure 3b), which could potentially arise from varying stratospheric and tropospheric air mass influences in different geographical regions. The lack of geographic variability contrasts with the reported regional $\Delta'^{17}O(CO_2)$ variability at the surface (Liang et al., 2017a, 2023). Apparently, the variability at the surface, caused by sources and sinks of $CO_2$, including isotope exchange with surface water reservoirs and anthropogenic

emissions, is no longer visible in the upper troposphere, where the reservoir seems to be relatively well mixed in terms of $\Delta'^{17}O(CO_2)$. Air masses in the upper troposphere experience rapid horizontal mixing, which can explain the uniform stratospheric influence in the CARIBIC samples.

The slope of two tracers near the tropopause is a measure of their relative net fluxes between troposphere and stratosphere (Plumb and Ko, 1992; Plumb, 2007). The slope of $N_2O$-$\Delta'^{17}O(CO_2)$ was first used by Luz et al. (1999) and Boering et al.

(2004) to quantify the so-called isoflux of $\Delta'^{17}O(CO_2)$ from the stratospheric into the troposphere. Previous measurements had a lower precision and could not define the slope close to the tropopause, but only for older stratospheric air. Our high precision measurements of $\Delta'^{17}O(CO_2)$ allow to determine the $N_2O$-$\Delta'^{17}O(CO_2)$ slope right down to the tropopause. Figure 8 shows





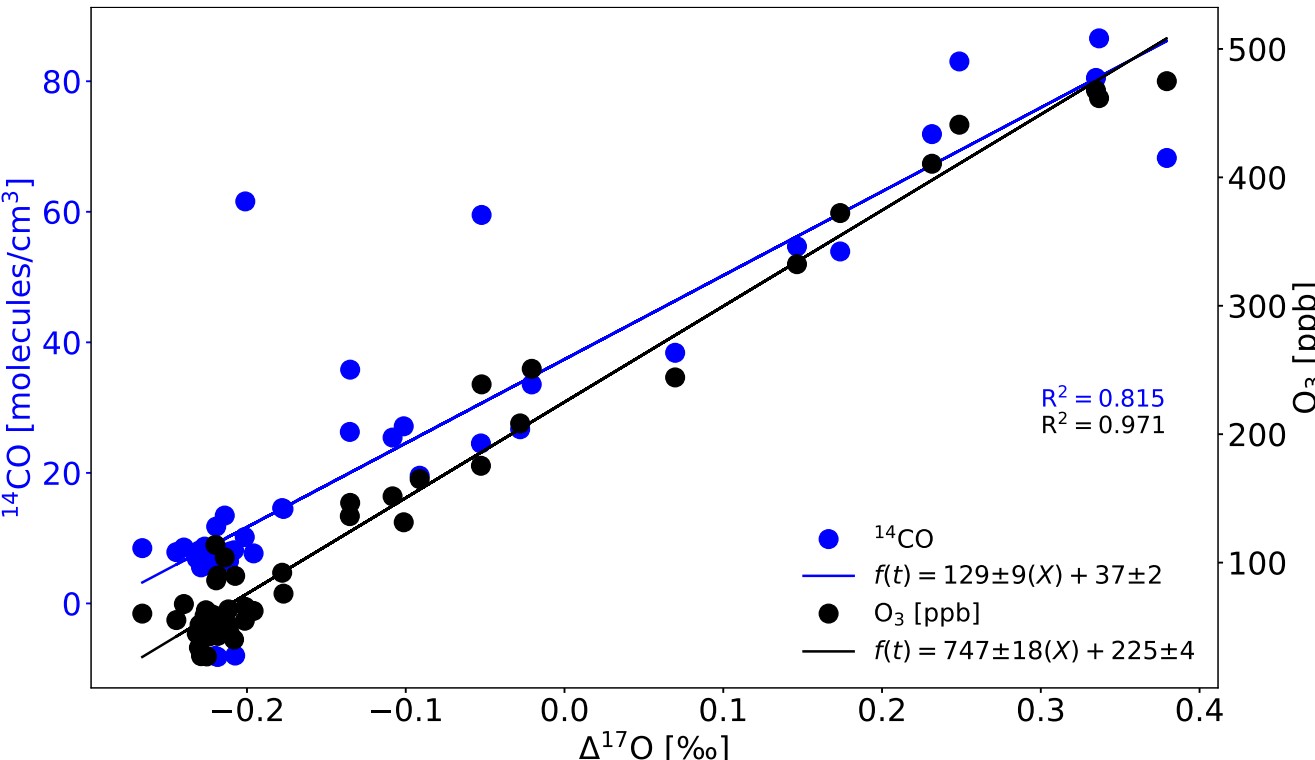

**Figure 7.** Correlation of $\Delta'^{17}O$ with stratospheric air tracers $^{14}CO$ and $O_3$ for CARIBIC samples.

that the slope for CARIBIC samples is indistinguishable from the correlations reported in the previous studies for stratospheric samples (-0.024 ‰/ppb) (Kawagucci et al., 2008; Wiegel et al., 2013; Boering et al., 2004). However, the regression lines

have an offset. This offset likely arises from variations in the $\Delta'^{17}O(CO_2)$ measurement scales among different laboratories. Recently, Liang et al. (2023) reported a scale offset in $\Delta'^{17}O(CO_2)$ of 0.037 to 0.042 ‰ between two laboratories (Institute of Earth Sciences, Academia Sinica and Institute of Earth Sciences, Hebrew University of Jerusalem). To optimize the application of $\Delta'^{17}O(CO_2)$ measurements, the implementation of interlaboratory calibration is crucial (Adnew and Röckmann, 2022). A recent study by Laskar et al. (2019) reported a higher slope (0.036 ‰/ppb) for the $N_2O$-$\Delta'^{17}O$ correlation of CARIBIC samples

collected during two flights. The samples they analyzed are mostly represent upper troposphere air, evident from a lower $O_3$ mole fraction compared to the CARIBIC samples measured in this study. This is supported by the relatively low three-isotope slope of 0.48 reported by Laskar et al. (2019) compared to the samples measured in this study (> 0.54). Furthermore, their samples cover only a very small range of $N_2O$ (only 3.5 ppb difference, i.e. between 311.5 and 315 ppb) which likely leads to a high error in the determination of the $N_2O$-$\Delta'^{17}O(CO_2)$ correlation slope.

In contrast, for the StratoClim samples, the $N_2O$-$\Delta'^{17}O(CO_2)$ correlation shows an unexpectedly low slope of -0.017 ‰/ppb, which is about 1.5 times lower than for the CARIBIC and other stratospheric samples (Figure 4). All our samples were



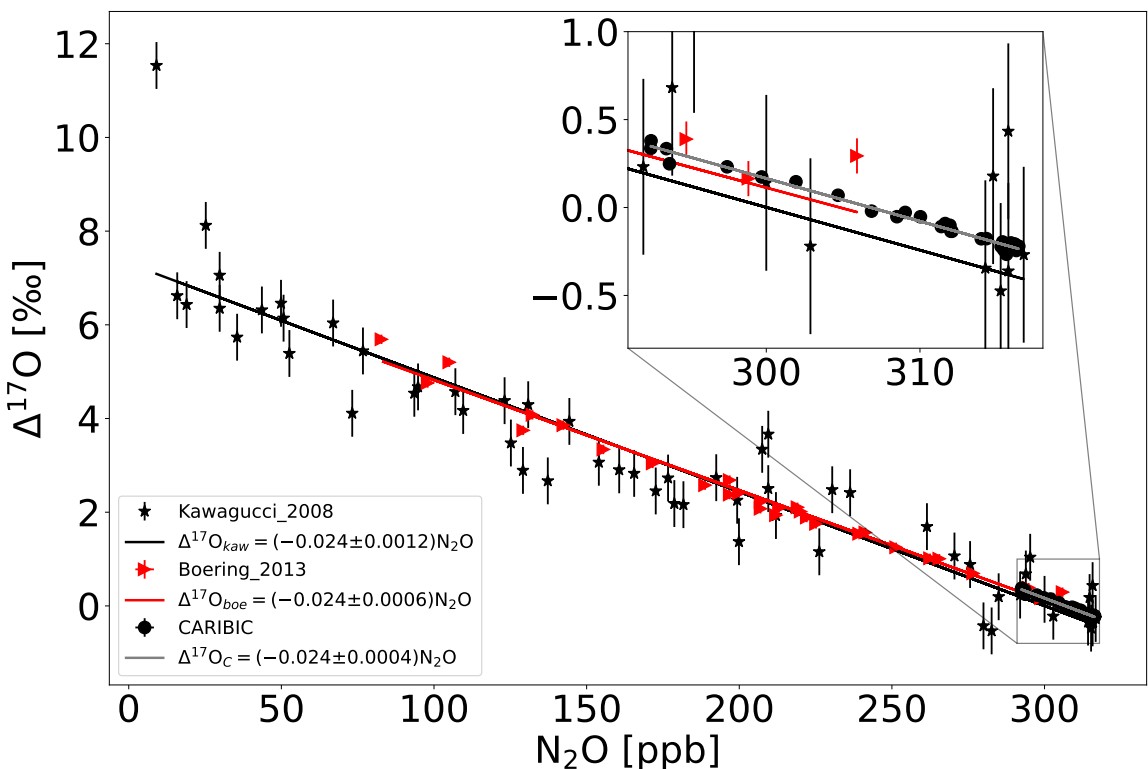

**Figure 8.** $\Delta'^{17}O - N_2O$ correlation for CARIBIC samples and measurements of Kawagucci et al. (2008) and Boering et al. (2004). The inset shows the zoom to the data close to the tropopause. For the CARIBIC samples measured in this study, the error-bars are smaller than the data markers. For the linear regression slopes, the errors are the 95 % confidence interval.

measured in the same laboratory and on the same analytical system. We suggest that the lower $N_2O$-$\Delta'^{17}O$ correlation slope for StratoClim samples is due to enhanced vertical mixing during the Asian summer monsoon anticyclone (ASMA), as these samples were collected during mid-June to early September. The ASMA causes deep convection and anticyclonic flow in the

upper troposphere and lower stratosphere (UTLS) where tropospheric air masses are uplifted into the UTLS (Park et al., 2009; Vogel et al., 2015, 2016; Brunamonti et al., 2018). The lifetime for $CO_2$ isotopic exchange is much slower than the transport time at all altitudes, whereas the photochemical production and quenching rates for $O^1(D)$ are much faster than transport processes. Therefore, the isotopic composition of stratospheric $CO_2$ should reflect both the variety of transport histories of air parcels and the sources of $O^1(D)$ (Liang et al., 2007; Boering et al., 2004). Furthermore, throughout the entire atmosphere, the

$CO_2$ isotopic exchange time is longer than the transport time. However, the lifetime of $N_2O$ varies with altitude. In the lower stratosphere, the lifetime of $N_2O$ against photolysis is longer than the transport time, resulting in less variability in the region.





However, in the middle and stratosphere the $N_2O$ life time decreases resulting a different $N_2O$-$\Delta'^{17}O$ correlation slope (Liang et al., 2008). The influence of vertical mixing on $N_2O$-$\Delta'^{17}O$ correlations was investigated in a model study by Liang et al. (2007). Indeed, they found that an increase in the vertical eddy diffusion coefficient - thus more vigorous vertical mixing -

leads to a lower slope of the $N_2O$-$\Delta'^{17}O$ correlation. This supports our hypothesis that the low correlation for the StratoClim samples is caused by intense vertical mixing in the ASMA. An independent indication for the enhanced mixing is the scatter observed in three isotope plot and air age variability for the samples classified as stratospheric (Figure 6). Enhanced mixing, probably caused by deep convection and anticyclone flow in the upper troposphere and lower stratosphere, is consistent with the enhanced mole fraction of ozone-depleting substances in the samples collected in the same flight (Adcock et al., 2021).

High-precision $N_2O$-$\Delta'^{17}O(CO_2)$ correlation can thus help identify enhanced mixing of tropospheric air into the stratosphere and maybe a measurable tracer to quantify the intensity of eddy diffusion/transport (Liang et al., 2007; Boering et al., 2004). There are many other indicators for enhanced mixing above the tropopause due to ASMA including a 25 % increase in long-lived ozone-depleting substances in the upper troposphere-lower stratosphere from the same samples (Adcock et al., 2021) and a contribution of about 30 % of young tropospheric air in the extratropical lower stratosphere in the Northern Hemi-

sphere due to ASMA (Vogel et al., 2016). Recently, Ma et al. (2022) reported that 30 % to 50 % of the air mass in the UTLS above the ASMA region is mixed, with the highest mixing occurring around 16.5 km altitude. Moreover, for StratoClim samples collected over the Indian subcontinent in Kathmandu (2017) in the tropopause region, the three major chlorinated very short-lived substances were enhanced up to 136 % compared to typical tropical tropopause values in 2013–2014. In contrast, only a 10 % increase was observed in ground-based measurements from 2014 to 2017 (Adcock et al., 2021; Engel et al., 2019).

For the StratoClim samples, the CO mole fraction reached a stratospheric background value at a potential temperature of 420 K, and the $N_2O$ mole fraction remained similar to the tropospheric value up to 400 K, as described in detail by von Hobe et al. (2021). The deviation of the correlation between $N_2O$ and $\Delta'^{17}O$ is observed for air samples where the potential temperature is higher, about 420 K and above (Figure 9). At these potential temperatures, mixing processes become more significant and the air inside the anticyclone is exported vertically and horizontally into the surrounding stratosphere (von Hobe et al., 2021;

Ma et al., 2022). This enhanced mixing apparently causes the diminished slope for the $N_2O$ and $\Delta'^{17}O$ correlation plot (Figure 4) and also a very scattered three isotope plot (Figure 6). Up to a potential temperature of 400 K to 415 K the strong isolation of air inside the ASMA prevents significant mixing of the stratospheric air into the predominately tropospheric inner cyclone (von Hobe et al., 2021). Consequently, the $N_2O$-$\Delta^{17}O$ correlation slope of the lowermost stratospheric StratoClim samples agrees with the ones for CARIBIC samples and previously published measurements.

## 365   5   Implication for troposphere and stratosphere exchange and surface emissions of $CO_2$

Air samples from the upper troposphere and lower stratosphere are well suited for studying stratosphere-troposphere exchange (Olsen et al., 2001). We used CARIBIC samples with $\Delta'^{17}O(CO_2) > -0.2$ ‰ , because for this group $\Delta'^{17}O(CO_2)$ and $N_2O$ have a very compact relationship with a slope of -0.024 ± 0.0002 ‰ ppb$^{-1}$, despite the air samples being collected at different latitudes, seasons, and years (see Table S1, Figure 1 and Figure 3). The $\Delta'^{17}O(CO_2)$: $N_2O$ correlation slope is similar to the



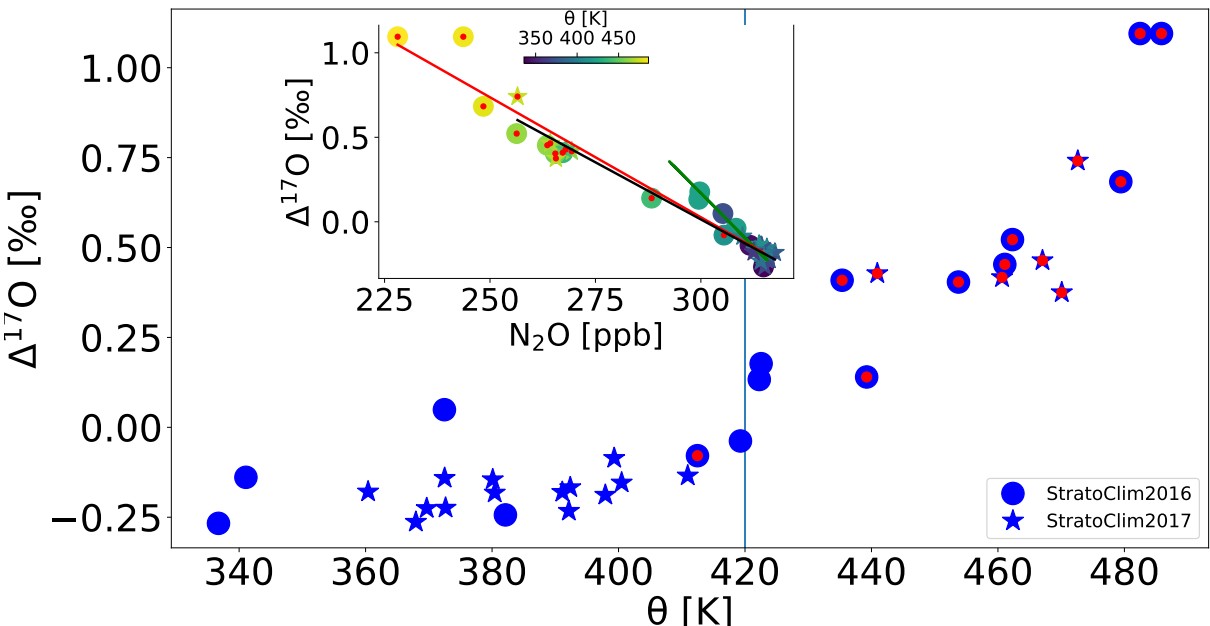

**Figure 9.** Correlation between $\Delta'^{17}O$ and potential temperature for StratoClim samples. The inset shows the $N_2O$-$\Delta'^{17}O$ correlation of StratoClim samples from Figure 4. The color bar for the inset is potential temperature. The data points marked in red indicate those with a low $N_2O$-$\Delta^{17}O$ correlation slope.

values reported in the previous measurements (Boering et al., 2004; Wiegel et al., 2013; Kawagucci et al., 2008). However, the uncertainty of the correlation slope is ten times smaller than in previous measurements (see Garofalo et al. (2019) for a detailed summary), and the new data confirm that the correlation actually extends down to the troposphere. Using the net vertical flux of $N_2O$ (which is equal to the global $N_2O$ loss rate, 13.43 TgNyr$^{-1}$ $\pm$ 3 % (Prather et al., 2023); see also (Tian et al., 2020)), the global mean net flux of $\Delta'^{17}O(CO_2)$ is 51.3 $\pm$ 1.6 ‰PgCyr$^{-1}$. This estimate aligns well with previous numerical model

calculations (Liang et al., 2008) and falls within the range reported previously using a similar approach (Boering et al., 2004; Garofalo et al., 2019; Wiegel et al., 2013; Kawagucci et al., 2008), but the uncertainty in our estimate is much lower than the 10 to 20 ‰PgCyr$^{-1}$ uncertainty reported in the previous studies (Garofalo et al., 2019). The relatively precise estimate of the net $\Delta'^{17}O(CO_2)$ flux from the stratosphere to the troposphere results from the improved precision of $\Delta'^{17}O(CO_2)$ measurements (see Figure 8) and the reduced uncertainty in the global $N_2O$ loss rate (3 % vs. 25 %).

Using the mass balance model described in section 2.3 (see equation 8), we estimated a terrestrial flux of 749 $\pm$ 93 PgC/year. This estimate falls within the range,near the upper range of values, reported by previous studies (200 - 817 PgC/year) (Ciais et al., 1997a, b; Cuntz et al., 2003b, a; Farquhar et al., 1993; Liang et al., 2017b; Welp, 2011). The surface flux, which is the sum of terrestrial and ocean fluxes, is 829 $\pm$ 93 PgC/year (see Figure 11). This value is consistent with the range reported by

**Figure 10.** a) The correlation of two short-lived gases, ozone vs CO for Kathmandu samples, b) Correlation of $\Delta'^{17}O$ with CO for both Kalamata and Katmandu samples. The data points marked in red indicate those with a low $N_2O$-$\Delta^{17}O$ correlation slope.

Welp et al. (2008) using $\delta^{18}O$ of $CO_2$. The corresponding surface turnover time of $CO_2$ is 0.98 ± 0.1 years, also within the range of previous estimates (i.e., 0.4 to 2.8 years) (Liang et al., 2017b, 2023; Laskar et al., 2019; Welp et al., 2011; Farquhar et al., 1993; Ciais et al., 1997a; Cuntz et al., 2003b) (see Figure 11). As shown in Figure 11, the estimated surface flux and turnover time of $CO_2$ are sensitive to the $\Delta'^{17}O(CO_2)$ value in the upper troposphere. A higher $\Delta'^{17}O(CO_2)$ value in the upper troposphere results in a lower surface flux and a longer $CO_2$ turnover time. A detailed sensitivity analysis is provided in the supplementary material. The GPP value is 211 ± 26 PgC/year, which is in good agreement with the estimates (200 PgC/year) by Liang et al. (2023) and Hofmann et al. (2017) using $\Delta'^{17}O(CO_2)$ and higher than the estimate (150-175 PgC/year) by Welp et al. (2008) using $\delta^{18}O$ of $CO_2$. These estimates are very sensitive to the assumed $c_m/c_a$ ratio, the degree of equilibration (Cuntz, 2011; Farquhar et al., 1993) and the soil invasion flux (Wingate et al., 2009). For example, increasing the $c_m/c_a$ ratio by 0.05 ($c_m/c_a$ = 0.7) decreases the GPP from 212 PgC/year to 189 PgC/year. Conversely, decreasing the $c_m/c_a$ ratio by 0.05 ($c_m/c_a$ = 0.6) will increase the GPP from 212 to 234 PgC/year. A detailed sensitivity analysis is provided in the supplementary material. Although the approach involves a number of assumptions as detailed in the methods section, it has the potential to be a valuable tool for quantifying and refining gross fluxes (Hoag et al., 2005; Koren et al., 2019). This will be especially true when more measurements become available from the lower stratosphere/upper troposphere and troposphere/surface. Additionally, understanding the effects of various processes on $\Delta'^{17}O(CO_2)$, such as soil invasion, and taking into account inter-laboratory





calibration scale differences (Adnew and Röckmann, 2022; Liang et al., 2023) and proper model assimilation (Koren et al.,
2019), will further enhance its effectiveness.

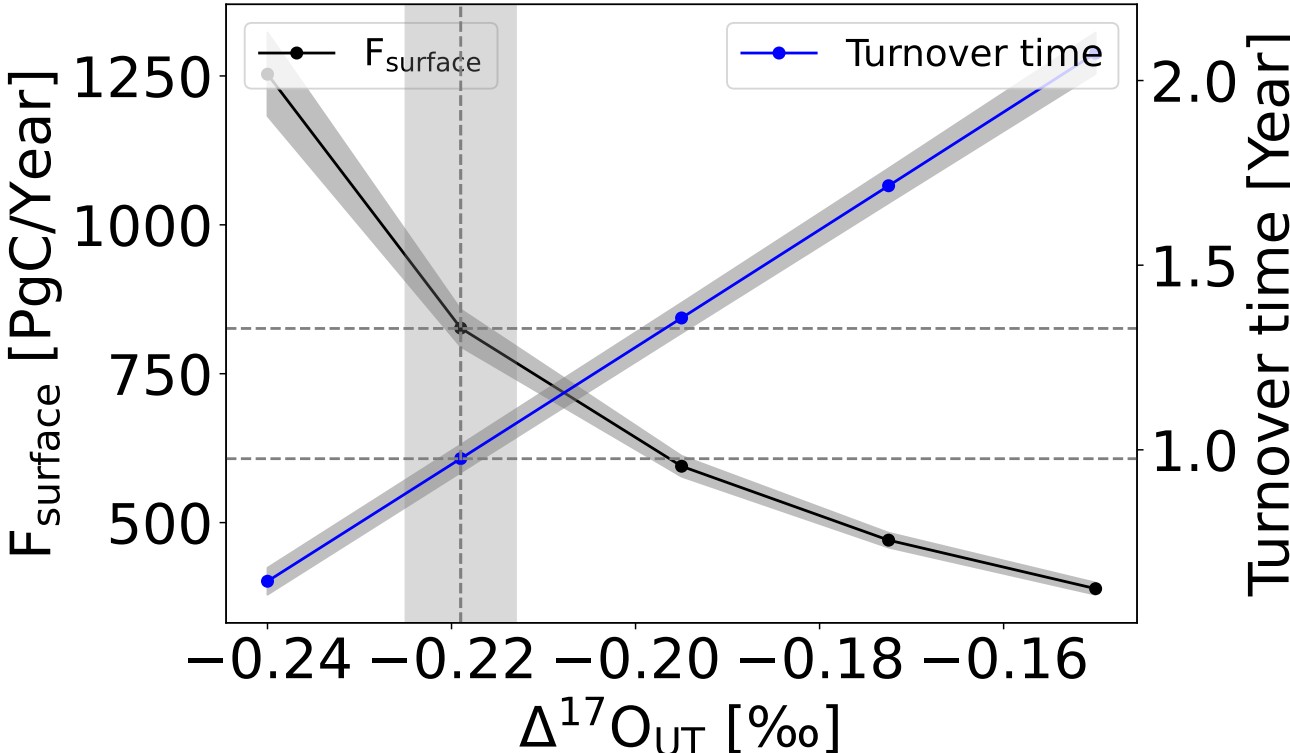

**Figure 11.** Surface flux of $CO_2$ ($F_{Surf}$, left axis) and the oxygen isotope turnover time of $CO_2$ (left axis) as a function of the $\Delta'^{17}O_{UT}$ value
of upper tropospheric $CO_2$. The solid line indicates our best estimates of $F_{Surf}$ and the oxygen isotope turnover time of $CO_2$ and the shaded
area is the corresponding uncertainty calculated using error propagation.

## 6 Conclusions

Through the analysis of CARIBIC samples, this study showed that the $N_2O$-$\Delta'^{17}O(CO_2)$ correlation reported previously from
stratospheric samples extends down to the tropopause. Using the $N_2O$-$\Delta'^{17}O(CO_2)$ correlation slope of CARIBIC samples,
we estimated the net mean $\Delta'^{17}O(CO_2)$ flux from the stratosphere to the troposphere to be $51.3 \pm 1.6$ ‰$PgCyr^{-1}$. Notably, no
significant spatial or hemispheric variability was observed in $\Delta'^{17}O(CO_2)$ values for the upper tropospheric samples collected
during the CARIBIC program. The new measurements can be used in a mass balance approach to estimate that the surface
turnover time of $CO_2$ is approximately 1 year, and the GPP is estimated to between 185 to 237 PgC/year.



In contrast, StratoClim samples showed a much lower $N_2O$-$\Delta'^{17}O$ slope compared to CARIBIC samples and previous studies. This deviation is attributed to the increased mixing/eddy diffusion due to ASMA. $N_2O$-$\Delta'^{17}O$ slope may thus be a
direct measurable tracer for the intensity of vertical mixing in the UTLS region.

*Data availability.* All the data is reported in the form of tables

*Author contributions.* GAA: Conceptualization; data curation; formal analysis; investigation; methodology; software, visualization; Writing - original draft; Writing - review & editing, GK:investigation, Writing - review & editing, NM: data curation, Writing - review & editing, SG:investigation, MK: data curation, Writing - review & editing, TR: Conceptualization; investigation; methodology, Writing - review &
editing

*Competing interests.* Some authors are members of the editorial board of Atmospheric Measurement Techniques. The authors declare no competing interest.

*Acknowledgements.* The StratoClim project was funded by the European Commission withiin Framework Prgram 7 under ENV.2013.6.1-2, Grant agreement No. 603557. The authors acknowledge Carl Brenninkmeijer and his team for providing the CARIBIC samples and
trace gas data. We also thank Carl Brenninkmeijer for his constructive feedback on the draft manuscript. Additionally, we appreciate the contributions of all staff involved in the regular maintenance of the IAGOS-CARIBIC container, flight preparation, handling of air sampling units, extraction of $CO_2$ from whole air samples, and analysis of trace gases used in this study. We are grateful to the staff who performed the sampling and trace gas analysis of StratoClim samples and provided us with the trace gas data. Furthermore, we acknowledge the technical support from Carina van der Veen, Marcel J. Portanger, and Henk Snellen. We acknowledge A. H. Laskar and M-C. Liang for constructive
discussion.



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
