# Peer review of "Triple oxygen isotope composition of $\mathrm{CO}_2$ in the upper troposphere and stratosphere"

_EGUsphere, 2024_

## Author Response (AR1)

From: Getachew Agmuas Adnew and co-authors

Anonymous Referee #1

We thank referee 1 for the constructive feedback and suggestions on how to revise the manuscript. The answers to the questions/ comments and suggestions are stated below each comment in blue. *Paragraphs that are modified in the revised manuscript are shown in blue (italics).*

This work presented high-precision measurement results of CO2 triple oxygen isotopes from upper troposphere lower stratosphere air samples up to 21 km collected during past aircraft campaigns. The results are interesting as it showed distinct relationship between triple oxygen isotopic compositions and N2O for air in the upper troposphere vs. lower stratosphere. Such observation is critical to enable CO2 triple oxygen isotopes as a tool to understand stratosphere-troposphere exchange, as well as global carbon cycle. This work highlighted the importance of high-precision triple oxygen isotopes measurements during quantification of the downward net isoflux of O-MIF signal. While the presentation of the results is clear, interpretation of the data is mostly adequate, I have a few minor general comments:

We appreciate your kind words about our work presented in the manuscript and your valuable comments.

1. It would be great if there is more discussion about the de-coupling of chemical mechanisms of CO2-O17 generation and N2O loss in the stratosphere due to stratospheric dynamics (Lines **325 - 344**). The CO2-O17 signal is originated from ozone chemistry, therefore the path history (O1D abundance vary greatly in the stratosphere) and age of the air parcel are both important; while N2O is more sensitive to altitude as the photochemical lifetime of N2O decrease exponentially in the stratosphere. Therefore, air parcels that are relatively "young" but have been to mid-stratosphere (~30 km) could have significant N2O loss but low O17, and vise versa. Clarifying some of these mechanisms could be useful.

In section 2.2 of the revised manuscript, we included the following paragraphs for clarification:

*The isotopic composition of $CO_2$ in the upper stratosphere and mesosphere provides a unique tool for studying atmospheric transport and chemistry [Wiegle et al. 2013; Liang et al., 2007, 2008, boering et al., 2004]. The $\Delta^{17}O$ of $CO_2$ is primarily modified by $O(^1D)$, which is produced photochemically by $O_3$ photolysis. However, the relevant isotope effects occurring in the stratosphere are still not yet well enough understood [Wiegle et al.2013; Liang et al., 2007, 2008]. Nevertheless, an empirical estimate of the isotope flux*

*from the stratosphere can be derived from measurements near the tropopause, like the ones presented here*

*In the stratosphere $CO_2$ and $N_2O$ isotopes are influenced by different processes. $N_2O$ is mainly destroyed by $N_2O$ photolysis but is also affected by $O(^1D)$ in the lower stratosphere and upper troposphere. Since $N_2O$ photolysis and $O_3$ photolysis occur at different wavelengths, the relationship between $\Delta^{17}O(CO_2)$ and $N_2O$ contains valuable information about atmospheric chemistry and transport. The lifetime of $N_2O$ varies with altitude. In the upper stratosphere and mesosphere, the $N_2O$ lifetime decreases, leading to greater scatter between the two tracers. $\Delta^{17}O(CO_2)$ values increase with altitude as $N_2O$ mixing ratios decrease below ~70 km. However, above 70 km, $\Delta^{17}O(CO_2)$ begins to decrease with further decreases in $N_2O$ mixing ratios. However, in the lower most stratosphere and upper troposphere, where the lifetime of $N_2O$ against photolysis is longer than the transport time, the scatter in $N_2O$ values remains low. The $\Delta^{17}O–N_2O$ correlation remains consistent both spatially and temporally in the lowermost stratosphere and upper troposphere (Liang et al., 2007, 2008). Since the net isotope flux of $\Delta^{17}O$ is derived from samples from the lower stratosphere and upper troposphere, the observed variability (scatter) in the stratosphere does not affect the global average $\Delta^{17}O–N_2O$ slope used to estimate the flux of $\Delta^{17}O$ from the stratosphere to the troposphere.*

2. More discussion may be needed to support the argument that the slope from CARIBIC samples can represent a "global average" N2O-O17 slope. Because of the observed potential "de-coupling" of N2O-O17 slope, it could be useful to discuss what are the potential factors that could result in different slopes. If the well-mixed upper trop air from CARIBIC represents global average slope, StratoClim gives you "below average" slope, where can you anticipate "higher than average" slopes? How will such variations impact the uncertainty of the global average slope?

   *See our reply to the previous comment.*

3. If the uncertainty in "global average" slope changed because of 1) and 2), how does it impact the uncertainties in global estimation of O17 isoflux?

*See previous comment*

More detailed comments:

Sections 2.2 & 2.3: since these were not mentioned until section 5, maybe considering moving these down (after 2.5) a little bit?

Thank you very much for your suggestion. In the revised manuscript we moved section 2.2 and 2.3 to appear after 2.5.

Line 174: maybe briefly mention how age of air is calculated?

In the revised manuscript we included a reference to the publication that was used to calculate the age of air.

*The age of air was calculated using SF6 measurements as described in detail by Krol et al., 2018.*

Line 249: CO2 not CH2.

Thank you very much for spotting this. In the revised manuscript the typo is corrected.

Figure 5: subpanel titles (a, b, c, d) are not lined up.

In the revised manuscript the subpanel title is aligned.

Figures 3-8: $\Delta'^{17}O$ is used in your text but in figures you used "$\Delta^{17}O$", please consider using consistent notations.

In the revised manuscript we have changed the $\Delta^{17}O$ in the figure label to $\Delta'^{17}O$.

Line 368: uncertainty inconsistent with figure.

Thank you very much for pointing out the inconsistency, in the revised manuscript the uncertainty indicator in the figures has been corrected to be consistent with the text.

Lines 388 & 394: repetitive sentences.

In the revised manuscript the repeated sentence has been deleted.

References

Wiegel, A. A., Cole, A. S., Hoag, K. J., Atlas, E. L., Schauffler, S. M., and Boering, K. A.: Unexpected variations in the triple oxygen isotope composition of stratospheric carbon dioxide, PNAS, 110, 17 680–17 685, https://doi.org/10.1073/pnas.1213082110, 2013.

Liang, M.-C., Blake, G. A., and Yung, Y. L.: Seasonal cycle of C16O16O, C16O17O, and C16O18O in the middle atmosphere: Implications for mesospheric dynamics and biogeochemical sources and sinks of CO2 , J. Geophys. Res. Atmos., 113, D12, 2008.

Liang, M.-C., Geoffrey, A. B., Brenton, R. L., and Yung, Y. L.: Oxygen isotopic composition of carbon dioxide in the middle atmosphere, PNAS, 104, 5, 2007.

Boering, K. A., Jackson, T., Hoag, K. J., Cole, A. S., Perri, M. J., Thiemens, M., and Atlas, E.: Observations of the anomalous oxygen isotopic composition of carbon dioxide in the lower stratosphere and the flux of the anomaly to the troposphere, Geophys. Res. Lett., 31, 2004.

Krol, M., De Bruine, M., Killaars, L., Ouwersloot, H., Pozzer, A., Yin, Y., Chevallier, F., Bousquet, P., Patra, P., Belikov, D., et al.: Age of air as a diagnostic for transport timescales in global models, Geoscientific Model Development, 11, 3109–3130, 2018.

Anonymous Referee #2

*We thank referee #2 for constructive feedback and suggestions on how to revise the manuscript. The answers to the questions/ comments and suggestions are stated below each comment in blue.* Paragraphs that are modified in the revised manuscript are shown in blue (italics).

This study presents triple oxygen isotope ($\Delta'^{17}O$) data of $CO_2$ of tropospheric and stratospheric samples derived from two aircraft campaigns. $\Delta'^{17}O$ is a novel isotope tracer that can provide an additional constraint on troposphere-stratosphere exchange and help to more accurately quantify exchange fluxes in the global carbon cycle. The authors show that $\Delta'^{17}O$-$N_2O$ relationship differs for upper tropospheric and lower stratospheric air. Using a mass balance model, they use their data to quantify global carbon fluxes. Their results demonstrate the potential of high-precision $\Delta'^{17}O(CO_2)$ measurements for quantifying and refining grow carbon fluxes and understanding mixing, transport and production processes at the troposphere-stratosphere boundary.

The manuscript is well-written, methods are described in detail, results are well-illustrated and key findings are highlighted in the discussion. I have only some very minor comments, as outlined below.

*We appreciate your acknowledgment of our work presented in the manuscript and your comments.*

**Minor comments:**

Line 119: "the box model used in this study" you refer to the study of Koren et al (2019) or to the present study? Please clarify. Also, can you briefly describe the model of Hoag et al (2005 and Liang et al (2017b) on which the model you used in this study is based?

*In the revised manuscript, we included the following description.*

*Hoag et al.(2005) used a two-box model to explore the relative contributions to the production of the $^{17}O$ anomaly in the stratosphere, its flux to the troposphere, and its destruction or dilution by various surface carbon fluxes in the biosphere. Liang et al. (2017) used a one-box model, in which major surface resetting processes were explicitly included to distinguish terrestrial (re-)cycling fluxes from oceanic fluxes.*

As described in the original manuscript, our model is an extended version of Hoag et al. (2005) and Liang et al. (2017).

**Line 149: which type of distribution did you use for the input parameters? Is it uniform or normal?**

We used a normal distribution for our input parameters. In the revised manuscript this has been specified.

Line 156: I found this sentence confusing. Consider reformulating to something like: To ensure comparability of previously published data and the CARIBIC and StratoClim samples measured in this study, ...

Thank you for your suggestion, the sentence has been reformulated in the revised version of the manusript accordingly.

Line 168/169: Can you explain briefly how this correlation allows differentiation between tropospheric and stratospheric samples?

In the revised manuscript we included the following paragraph:

*$N_2O$ maintains a nearly constant tropospheric concentration, as it is minimally affected by chemical processes and has no significant atmospheric sources. This stability allows stratospheric influence to be more easily identified. The correlation between $N_2O$ and CO forms an L-shaped curve, similar to $O_3$-CO correlations observed during stratosphere-to-troposphere airmass mixing events. As a result, the troposphere corresponds to the horizontal branch (high $N_2O$, variable CO), while the stratosphere, free from tropospheric influence, corresponds to the vertical branch (low CO, variable $N_2O$).*

Section 2.5-2.7: I suggest following the chronological order and presenting the sampling and isotope analysis of the samples used in this study before presenting the data-processing, calculations and models applied, where these are used.

In the revised manuscript, we have implemented the suggested chronological order.

Figure 3: I was confused by the illustration as histograms. First, I was thinking the height reflects the number of samples. You may think about illustrating the data just a points with SE?

Thank you for your suggestion, in the revised manuscript we have implemented the suggestion accordingly.

Line 269 ff: I suggest presenting data related to Figure 6 before Figure 5. Like this, you first present correlations found in your study before comparing both the triple oxygen isotope plot as well as correlations between N2O-Δ'17O and CH4-Δ'17O with previously published data.

In the revised manuscript, we present Figure 6 before Figure 5.

Conclusions: It would be great if the authors could highlight some key findings on the application of Δ'17O(CO2) here. I refer here mainly to the ability to identify mixing, transport and production processes at the troposphere-stratosphere boundary.

In the revised manuscript, we addeed the following sentence:

*High precision measurements of Δ'17O(CO2) enable identify mixing, transport and production processes in the stratosphere.*

**Technical comments:**

Line 114: doubling "depends on the uncertainty". Remove one.

Done

Line 118: change point to comma after "(see Figure 1)".

Done, comma to point instead.

Line 210: the third and "fourth" traps.

In the revised manuscript the suggestion is implemneted.

Line 320: Remove "are". "The samples they analyzed are mostly represent upper troposphere air"

Done

Line 137: Repetition "however" two times in a row. Consider changing to "in contrast". Also in this line: "In the middle and stratosphere". Here seems to be missing a word.

Line 337: Thank you for your suggestion. In the revised manuscript the sugggestion is implemented.

"In the middle and stratosphere" -> in the middle stratosphere

References

Hoag, K. J., Still, C. J., and Fung, I. Y.: Triple oxygen isotope composition of tropospheric carbon dioxide as a tracer of terrestrial gross carbon fluxes, Geophys. Res. Lett., 32, https://doi.org/10.1029/2004gl021011, 2005.

Liang, M.-C., Mahata, S., Laskar, A. H., Thiemens, M. H., and Newman, S.: Oxygen isotope anomaly in tropospheric CO2 and implications for CO2 residence time in the atmosphere and gross primary productivity, Sci Rep, 7, 13 180,https://doi.org/10.1038/s41598-017-12774-w, 2017.